# Regulation of Epithelial-Mesenchymal Transitions by Alternative Splicing: Potential New Area for Cancer Therapeutics

**DOI:** 10.3390/genes14112001

**Published:** 2023-10-26

**Authors:** Ling Li, Jinxia Zheng, Sebastian Oltean

**Affiliations:** Department of Clinical and Biomedical Sciences, Faculty of Health and Life Sciences, University of Exeter Medical School, Exeter EX1 2LU, UK; l.li3@exeter.ac.uk (L.L.);

**Keywords:** epithelial-mesenchymal transitions, alternative splicing, cancer

## Abstract

The epithelial-mesenchymal transition (EMT) is a complicated biological process in which cells with epithelial phenotype are transformed into mesenchymal cells with loss of cell polarity and cell–cell adhesion and gain of the ability to migrate. EMT and the reverse mesenchymal-epithelial transitions (METs) are present during cancer progression and metastasis. Using the dynamic switch between EMT and MET, tumour cells can migrate to neighbouring organs or metastasize in the distance and develop resistance to traditional chemotherapy and targeted drug treatments. Growing evidence shows that reversing or inhibiting EMT may be an advantageous approach for suppressing the migration of tumour cells or distant metastasis. Among different levels of modulation of EMT, alternative splicing (AS) plays an important role. An in-depth understanding of the role of AS and EMT in cancer is not only helpful to better understand the occurrence and regulation of EMT in cancer progression, but also may provide new therapeutic strategies. This review will present and discuss various splice variants and splicing factors that have been shown to play a crucial role in EMT.

## 1. Introduction

The epithelial-mesenchymal transition (EMT) is an important process in both normal embryonic development and wound healing, enabling the synthesis of human tissues and organs by converting cells from the epithelial phenotype into the mesenchymal phenotype [1]. However, EMT has also been found to be involved in the invasion and metastasis of tumours, among other important features of carcinogenesis [2]. More than 90% of all malignant tumours originate from epithelial cells, and cancer cells are able to spread to distant organs and infiltrate nearby tissues through the dysregulation of EMT [1,3]. EMT is therefore one of the primary drivers in the process of development of epithelial tumours into more aggressive phases. EMT is regulated at various levels, including transcriptional control, post-translational, differential splicing, and non-RNA regulation [4]. An essential component in the modulation of EMT is the transcriptional regulation, which is mediated by a group of transcription factors (TFs), including members of the SNAIL, TWIST, and ZEB families, that induce the expression of genes necessary for mesenchymal properties and inhibit the expression of epithelial-related genes [5]. However, accumulating evidence shows that alternative splicing also has an essential role in the modulation of EMT.

The process known as alternative splicing (AS), which produces numerous mRNAs from a single transcript, is one of the main factors that drives the variety of the proteome. During the development of cancer, the dysregulation of AS contributes to many aspects of the tumour cell biology, including EMT. A number of EMT-related molecules have been shown to be able to be regulated by multiple kinds of cancer-specific AS isoforms, which consequently promote the EMT process [6]. Hence, mounting evidence suggests that regulation of AS might be essential for multiple key characteristic alterations related to EMT. Understanding the EMT-related AS regulators and their resulting isoforms is crucial to comprehending how regulation of EMT through switching AS is a useful new perspective for cancer therapy.

In this review, we present a brief summary of the EMT process as well as evaluate the role of AS in EMT by providing examples of target genes and splicing regulators switching AS and regulating EMT in cancer. We also present ideas on how the manipulation of AS may be used therapeutically to reverse EMT and therefore slow tumour growth and metastasis.

## 2. The Epithelial-Mesenchymal Transition

EMT is a complex biological process in which cells with epithelial phenotype are transformed into mesenchymal phenotype with the loss of cell polarity and cell–cell adhesion and the gain of the ability to migrate [1]. The epithelial tissue is a protective barrier that covers the internal surface of organs and external body surface. Epithelial cells are closely adjoined by adherens junctions, tight junctions, and gap junctions to form cell layers. Mesenchymal cells can migrate and differentiate into other cell types without forming an organized layer of cells. Due to the inherent plasticity of the epithelial phenotype, EMT is reversible, and through mesenchymal-epithelial transformation (MET), epithelial phenotype may be restored. EMT is important for many developmental processes, including embryonic development and neural tube formation. EMT is present physiologically in embryogenesis and wound healing, but it is abnormally activated in chronic fibrosis, and in the initiation of metastasis in cancer progression. Research into the mechanisms of EMT and MET in tumourigenesis and metastasis has become a hot spot in cancer therapy.

### 2.1. Mechanism of EMT

The main features of EMT are downregulation of epithelial cell characteristics and acquisition of mesenchymal properties (Figure 1). The first step of EMT is a disassembly of the epithelial cell–cell contacts, which include the tight junction, adherens junction, desmosomes, and gap junction, as well as the loss of cell polarity. The key cell–cell adhesion molecules—epithelial cadherin (E-cadherin) and cytokeratin—are downregulated while neural cadherin (N-cadherin), vimentin, and fibronectin, are upregulated. Cells acquire the mesenchymal phenotype with motility and invasive capacities by forming lamellipodia, filopodia, and invadopodia, followed by expressing matrix metalloproteinases which have the ability to degrade extracellular matrix proteins [7].

### 2.2. Classification of EMT

EMT are categorized into three main groups based on functional characteristics:

Type 1 EMT (Figure 2A) is encountered during embryogenesis. The primitive epithelium transforms to the primary mesenchyme through EMT, then the primary mesenchyme is re-induced to generate secondary epithelia through the reverse MET. These secondary epithelia can then develop into diverse kinds of epithelial tissues. It is related to embryonic development, such as implantation, embryonic gastrulation, and organ development. During the embryogenesis, both EMT and MET can exist simultaneously for the differentiation of specific cells and the formation of complex structures of internal organs.

Type 2 EMT (Figure 2B) has been correlated to repair mechanisms that include wound healing, tissue regeneration, and organ fibrosis [9]. When tissues are chronically damaged, Type 2 EMT can stimulate a large number of fibroblasts to persistently secrete a large amount of collagen, resulting in tissue and organ fibrosis. This pathological mechanism is common in the development of many fibrosis diseases, such as liver fibrosis, idiopathic pulmonary interstitial fibrosis, and renal fibrosis.

Type 3 EMT (Figure 2C) is related to cancer invasion and metastasis, and is crucial for the ability of cancer cells to cross the basement membrane, enter the blood circulation, and colonize target organs [9].

### 2.3. Regulation of EMT

The mechanisms regulating EMT are quite complex. Mechanical signals and biochemical signals are both playing important roles in this regulation. There are many regulators at several different molecular levels, including transcriptional and post-transcriptional levels like mRNA processing or microRNAs [1].

Snail family zinc-finger transcription factor (SNAIL) is one of the most important transcription factors (TFs) involved in EMT; it was first found as a transcriptional repressor of E-cadherin and originally identified as a repressor of E-cadherin homologs in Drosophila [10]. Proteins within the SNAIL family in vertebrates have been subdivided into two subfamilies: Snail1 (known as SNAI1) and Slug (known as SNAI2) [11]. They both can induce EMT by directly inhibiting the transcription of E-cadherin [12]. Moreover, SNAIL also completely inhibits the integral membrane proteins, claudins, and occludin which disrupts the tight junction and promotes EMT [13].

Twist family BHLH transcription factor (TWIST) is a basic helix-loop-helix protein which is an important TF for EMT during metastasis and embryonic morphogenesis [14]. TWIST represses E-cadherin and induces EMT and cell migration by directly or indirectly suppressing the E-cadherin transcription via target E-boxes in the E-cadherin promoter [15].

Zinc-finger E-box binding homeobox (ZEB) family of TFs, similar to SNAIL and TWIST, bind the E-boxes and repress epithelial junctions and polarity genes as well as activate mesenchymal genes during EMT [16]. Two subfamilies of proteins from the ZEB family in vertebrates have been reported: ZEB1 and ZEB2. They can both inhibit or stimulate the transcription process by attaching to E-boxes on regulatory gene sequences [17].

MicroRNAs (miRNAs) are also key regulators of epithelial phenotype during EMT. They can regulate E-cadherin expression either directly or indirectly and regulate the expression of TFs during EMT, as well as other target genes, to assist in identifying the epithelial or mesenchymal phenotype [18]. Additionally, long-noncoding RNA (lncRNAs) as a post-transcriptional regulator are also involved in EMT regulation, while miRNAs regulate EMT-TF [18,19,20].

Among many cytokines that regulate EMT, numerous signalling pathways, such as transforming growth factor-β (TGF-β), Wnt, Notch, and Hedgehog are also involved in the regulation of EMT [21]. TGF-β has been proved to be a master regulator and primary inducer of EMT [22]. TGF-β is essential for carcinomas to gain the invasive phenotype and has been considered as a promoter of metastasis [23].

## 3. Alternative Splicing

EMT is regulated at several gene expression levels, one of them being alternative splicing (AS). AS is a form of regulated splicing, in which specific exons (or parts of them) are differentially skipped or included, resulting in various forms of mature messenger RNAs (mRNAs). More than 94% of human genes experience AS, which greatly expands the diversity of the transcriptome [24,25]. It is a major driving factor for proteomic diversity and gene regulation. The aberrant regulation of AS triggers several human diseases, and the deregulation of AS is a hallmark of cancer. Hence, understanding the mechanism and modulation of AS is important for the identification of new strategies for cancer therapy.

### 3.1. Splicing

The basic process of splicing contains two major steps (Figure 3): spliceosome assembly and pre-mRNAs actual splicing. These two processes are catalysed by the cooperation of five core small nuclear ribonucleoprotein (snRNP) particles (U1, U2, U4, U5 and U6) and numerous auxiliary proteins. The spliceosome is a macromolecular ribonucleoprotein complex composed of many proteins and small nuclear RNA molecules [26]. The process of spliceosome assembly is directed by consensus sequences at splice sites and results in the sequential binding and release of snRNPs, protein factors, and the disruption and formation of RNA–RNA, RNA–protein, and protein–protein interactions. Pre-mRNA splicing is a process of removing introns from pre-mRNA through two consecutive transesterification reactions and the ligation of exons in the mRNA [27].

### 3.2. Basic Modes of Alternative Splicing

There are five major types (Figure 4) [28]:Cassette exons (Exon skipping)—the most prevalent pattern in which exons are spliced out of the gene or retained in the transcript.Mutually exclusive exons—only one of two consecutive exons is retained in the mature transcript.Alternative 3′ splice site—when the splice junction at the 3′ end is changed.Alternative 5′ splice site—when the splice junction at the 5′ end is changed. Both alternative 3′ and 5′ splice sites can cause changes in the coding sequence.Intron retention—when an intron is retained in the final transcript.

### 3.3. Regulation of Alternative Splicing

The regulation of AS is a highly dynamic combinatorial process that relies on complex coordination between intracellular and post-transcriptional processes [29]. Regulation of AS can be triggered by the interaction of multiple RNA-binding proteins (RBPs), which bind to *cis*-regulatory elements around the splice sites resulting in the utilization of the regulated splice site being enhanced or repressed [26]. The *cis*-elements, which can be located in either exons or introns, are bound by regulatory proteins to enhance or silence splicing of adjacent regulated exons, to therefore determine whether the mature transcript includes or skips certain exons [30]. Splicing regulatory elements can be divided into four categories: exonic/intronic splicing enhancer/silencers [31]. Precise control of AS is vital for normal cells, as aberrant expression of splicing is a common cause of diseases including cancer. The trans-acting splicing factors belong mostly to two large families—SR proteins, which normally bind to splicing enhancers and promote spliceosomes assembly, and hnRNPs opposing SR protein function, which generally bind to splicing silencers to inhibit exon recognition and promote exon skipping [32,33].

Two of these splicing factors are most frequently associated with EMT: ESRPs and RbFox2. Epithelial specific regulatory proteins (ESRPs) are thought to be master regulators of EMT (there are two paralogues—ESRP1 and 2). ESRPs control the splicing of many gene transcripts during EMT, including FGFR2, CD44, ENAH. The ESRPs downregulation drives cells towards a mesenchymal phenotype [34]. RNA-binding protein FOX2 homologue (RBFOX2) contributes to EMT and cell invasion and is thought to regulate a splicing programme that defines mesenchymal states [35]. Ectopic expression of ESRP1 or depletion of RBFOX partially induced the epithelial splicing program in mesenchymal cells, conferring epithelial characteristics, which demonstrate a critical function for alternative splicing during MET [36].

## 4. Manipulation of Alternative Splicing in EMT as a Potential Therapy for Cancer

There is growing evidence that EMT and MET transitions play a crucial role in cancer progression [4,37,38,39,40,41,42,43,44,45]. Inhibition of EMT has become a very promising treatment avenue. However, the complexity of the signal pathways that regulate the EMT process, combined with the existence of a MET conversion mechanism that reverses the results of an EMT, makes choosing a target even more complicated. Among the different levels of EMT regulation—transcription, translation, post-translational modifications, alternative splicing, and non-coding RNA regulation—AS plays an essential role. There are many AS events involves in EMT during cancer progression, such as splicing of FGFR2, CD44, CTNND1, and ENAH and they may become therapeutic points- it is therefore timely to ask ourselves: Can these AS events in EMT be targets of cancer therapy? Can we try to manipulate AS in EMT to control cancer progression? Will using small molecules to switch splicing in EMT, or antisense oligonucleotides (ASOs), or antibodies recognizing cancer-specific splicing isoforms in EMT provide potential therapeutic strategies in cancer treatment? Will inhibitors targeting EMT-associated splicing events, splicing factors, or signalling pathways involving AS have great potential in cancer diagnosis and treatment?

### 4.1. Switching Specific Alternative Splicing Patterns in EMT

There are a series of alternative spliced events in EMT which play essential roles during cancer progression. Previous studies show that changing the splicing patterns of these gene isoforms can change phenotypes in both EMT and cancer. Thus, targeting splicing patterns is a potential method of cancer management.

#### 4.1.1. Fibroblast Growth Factor Receptor 2 (FGFR2)

FGFR2 is one of the well-recognized genes spliced differently in EMT which is essential in embryogenesis and organ regeneration [46,47,48,49]. The FGFR family functions through the binding of FGF ligands to their receptors, activation via dimerization and, therefore, prompting the tyrosine kinase domains to a set of subsequent signal activations [50]. Following this cell division, growth and differentiation are induced [51,52,53,54]. There are two mutually exclusive FGFR2 splice isoforms—FGFR2 IIIb and FGFR2 IIIc. Splicing switch from FGFR2 IIIb to FGFR2 IIIc has been implicated in pathological EMT [55]. Epithelial phenotype involves exon IIIb inclusion while, in EMT, exon IIIc inclusion induces a mesenchymal phenotype. It has been shown that ESRP1 and 2 are regulating splicing of FGFR2, switching to the epithelial phenotype—FGFR2 IIIb isoform [56]. ESRP expression is higher in epithelial cells and decreases in EMT. The switch from exon IIIc to IIIb of FGFR2 in mesenchymal cells may be promoted by ESRP overexpression. It was also found that the RNA-binding proteins hnRNPA1 and PTBP1/2, which are highly expressed in tumours, play a role in exon IIIc skipping [57]. Thus, FGFR2 splicing could be used as one of the targets for anticancer drugs development. Indeed, in a recent study from our lab [58] we performed a screen using FGFR2-based splicing reporters and have identified small molecules that switch FGFR2 splicing and EMT phenotypes; therefore, modulation of FGFR2 splicing could be a potential cancer therapeutic target.

#### 4.1.2. Recepteur d’Origine Nantais (RON, MST1R)

The proto-oncogene RON, also known as macrophage stimulating 1 receptor (MST1R), is a member of the receptor tyrosine kinase (RTK) family. The mature RON protein is a 180 kDa heterodimer composed of a 40 kDa α chain and a 150 kDa β chain. The transmembrane β chain has tyrosine kinase activity, and the precursor (pro-RON) exists as a single chain [6]. The extracellular sequence of RON includes amino-terminal semaphoring (Sema), plexin-semaphorin-integrin (PSI), and four immunoglobulin-like IPT domains. Variants are mainly generated through mechanisms such as alternating pre-mRNA splicing, protein truncation and alternative transcription. RON and its alternatively spliced variants, including RONΔ85, Δ110, Δ155, Δ160, Δ165, and Δ170, play essential roles in numerous tumour biological activities in cancer, such as cell–cell adhesion, proliferation, apoptosis, and EMT [59]. The production of the ΔRon165 isoform during EMT is generated by exclusion of exon 11, which results in lacks of 49 amino acids (aas) in the extracellular β-chain [6]. The inclusion or exclusion of exon 11 is regulated by two neighbour regulatory elements, an ESS and an ESE, which are located in the downstream exon 12 [60]. Mayer et al. revealed that RONΔ165 were detected three times more frequently in tumour samples (82.22%) in comparison to the alternatively spliced RON variant with exon 5/6 exclusion (potential RONΔ160 or RONΔ155) (24.40%) [61]. Assorted RON protein isoforms produced by spliced variants play different roles in tumour cell properties both in vitro and in vivo. The novel RONΔ165 variant stimulated tumour progression while initiating the PI3K/AKT pathway via PTEN phosphorylation [62]. Gupta et al. also found that hnRNPA2B1 induce the skipping of exon 11 caused generation of RONΔ165, which eventually induce EMT through activation of Akt/PKB signalling in head and neck cancer [63]. RONΔ160, an oncogenic protein variant of RON, triggers structural alterations that induce cellular transformation in vitro and tumour growth in vivo in human colon cancers [64,65]. RONΔ155, generated from a combined deletion of exons 5, 6, and 11, has similar functions to RONΔ160 [66,67]. Another variant, RONΔ170, is produced by exclusion of exon 19 in AS which inhibits biological processes involving tumour initiation intermediated by oncogenic variant RONΔ160 in colon cancer cells [68]. RON splicing is also involved in the occurrence and development of lung, colon, and breast cancers, suggesting that it can be used as a new prognostic indicator and treatment target for various cancers [64,69,70,71]; for example, RONΔ155 and RONΔ160 could be used as indicators of tumour growth and the switch from RONΔ160 or RONΔ165 to RONΔ170 may help with the inhibition of tumour initiation.

#### 4.1.3. CD44

The protein encoded by CD44 is part of a group of integral membrane glycoproteins which mediate cell–cell interactions and the interaction between cells and the extracellular matrix, together with cell adhesion and migration. The CD44 gene transcript undergoes complex alternative splicing, resulting in many functionally different protein subtypes. These different isoforms have diverse tissue-specific functions and are involved in various cellular processes, including tumour progression and metastasis. It has been found that expression of CD44v (CD44 novel splice variant) isoforms is often associated with initiation, progression, and metastasis of colon, prostate, intestinal, gastric, and breast cancers [72,73,74,75,76,77]. Zhang et al. uncovered that in breast cancer cells CD44 could switch between CD44s (CD44 standard splice variant) and CD44v to gain different properties like proliferation and migration. CD44s help cancer cells gain stem cell properties, promoting tumour metastasis or reoccurrence, while cells with more CD44v isoform are highly likely to have less cancer stem cell properties and a higher proliferation rate [77]. Cheng et al. reported CD44s expression was increased in high-grade human breast tumours and was associated with N-cadherin increase in these tumours [78]. It might be able to change the cancer cell properties and therefore improve the sensitivity of cancer cells to treatment by manipulating the ratio of these two types of isoforms. In prostate cancer cells, CD44v has a higher expression level in epithelial cells in comparison to mesenchymal cells [79], which is consistent with the previous research indicating CD44v can suppress EMT [80].

#### 4.1.4. Catenin Delta 1 (CTNND1)

The CTNND1 gene encodes a member of the Armadillo protein family, also known as p120, which plays a role in both oncogenic and tumour suppressor functions, and its alternative splicing is related to cell proliferation, migration, invasion, and EMT [81,82,83]. Two isoforms of CTNND1 generated by AS, p120-1 and p120-3, are associated with distinctive functions and different interactions in various cell types. A long mesenchymal-specific splice variant with exons 2 and 3, p120-1 is normally mostly expressed in mesenchymal cells, whereas p120-3 is a shorter isoform with lack of these exons and is mostly predominant in epithelial cells [84,85]. p120 isoform expression evaluation within a panel of breast cancer cell lines revealed that more invasive cells show higher expression of the large isoforms p120-1 and p120-2; however, p120-3 is expressed in all cell lines [86]. As p120-3 often has high expression levels in epithelial cells, a switch from isoform p120-1 to p120-3 might be a possible way to revise EMT and inhibit tumour growth.

#### 4.1.5. Other Genes Spliced Differently during EMT

There are many other genes that are alternatively spliced during EMT and play essential roles in tumour initiation and progression.

**ARHGEF11** (Rho guanine nucleotide exchange factor 11) is the guanine nucleotide exchange factor (GEF) for the RhoA small GTPase protein. It has been revealed to promote tumour metastasis in glioblastoma and ovarian carcinoma and promotes proliferation and EMT of hepatocellular carcinoma by activating β-catenin [87]. Previous studies show that the expression of the ARHGEF11 isoform containing exon 38 is correlated to the malignant phenotype in breast tumours, which means ARHGEF11 exon 38(+) could be used as a biomarker and target of breast cancer management [88].

**CCND1** has two isoforms derived by splicing, named cyclin D1a and cyclin D1b, with inclusion of intron 4 in D1b mRNA. Both D1a and D1b are often upregulated in human cancers; however, cyclin D1b alone can induce cellular transformation, and is correlated with cancer progression and poor prognosis in prostate, colon, colorectal, and urinary bladder cancer [89,90,91,92,93].

**Androgen receptor** (AR) variants, especially AR3, could contribute to prostate cancer progression through inducing EMT, achieving stem cell characteristics, and regulating stem cell related pathways [94,95,96].

Research revealed that the expression of **MBNL1** isoforms lacking exon 7 (MBNL1 Δex7) proteins plays a role as a tumour suppressor, as cancer cells tend to downregulate in the presence of MBNL1 isoforms containing exon 7 [97].

**Zinc-finger antiviral protein** (ZAP) is an important antiviral factor that specifically inhibits the replication of a variety of viruses by binding to the target RNA sequence of the virus and interfering with the translation initiation of the target mRNA. ZAP has two major isoforms that result from alternative splicing at their C terminus: ZAPL (long) encodes a poly (ADP-ribose) polymerase (PARP)-like domain that is missing from ZAPS (short). Recently, it was reported that ZAPL promotes EMT while ZAPS suppresses it in HEK293T cells while ZAPS promotes migration in MCF-7 cells. However, ZAPL and ZAPS play different roles at different stages of EMT in human breast cancer cells [98].

### 4.2. Targeting Splicing Regulators Involved in EMT

Several splicing factors have been reported to be involved in regulating EMT. Modulating their activity could be another therapeutic strategy.

#### 4.2.1. Epithelial Splicing Regulatory Proteins (ESRPs)

ESRPs drive splicing events of about 200 genes, including FGFR2, NUMB, EXOC1, MAPK14, SCRIB, and GSK3, switching these genes splicing towards isoforms characteristic of the epithelial phenotype [99,100]. It has been shown that ESRPs play important roles in regulating EMT and are crucial in maintaining epithelial properties and therefore reducing tumour transformation towards cells with mesenchymal phenotype [99]. In EMT, ESRPs are downregulated, and their targets switch splicing to the mesenchymal phenotype. ESRPs have dual roles in tumour progression—either downregulated or upregulated. ESRPs are low in normal epithelial cells but higher in primary and advanced tumours. They decrease in expression in regions before the invasion areas in tumours [101]. It has been reported that ESRP1 and 2 reduce mobility by downregulating Rac1b and δEF1/SIP1 expression, respectively, and decrease proliferation of tumour cells [101,102]. One possibility is ESRP2 inhibits δEF1 and SIP1 by controlling miRNA expression. It is reported that the overlap of pre-miRNA sequences and active splice sites triggers competition between alternative splicing mechanisms and miRNA processing machinery [103]. Furthermore, it has been shown that overexpression of ESRPs significantly decreases prostate tumour growth in xenografts mouse models [104]. Therefore, manipulating ESRPs expression level will hopefully become a new way to contribute to the management of cancer.

#### 4.2.2. RNA-Binding Fox Protein 2 (RBFOX2)

RBFOX2 contains an evolutionarily highly conserved RNA recognition structural motif (RNA recognition motif, RRM), which can specifically bind to the (U)GCAUG sequence. The activation or inhibition of splicing by RBFOX proteins depends on the location of their binding sites. The binding sites are located upstream of the exons. RBFOX proteins can inhibit splicing, but when located downstream of the exons, they usually activate splicing. Several studies implicated the splicing factor of RBFOX2 in splicing regulation in EMT during cancer progression [35,36,105]. Interestingly, RBFOX2 is also involved in the regulation of FGFR2 splicing. The function of RBFOX2 in EMT is controversial as RBFOX2 regulates both epithelial and mesenchymal splicing events, which might be correlated to the binding site’s location. EMT induces RBFOX2 upregulation while reversing EMT restores the expression levels of RBFOX2. RBFOX2 expression and splicing activity are controlled by EMT transcription factors [36]. However, Braeutigam et al. suggest that during EMT, RBFOX2 drives a mesenchymal tissue-specific splicing program, contributing to cell invasion [35,105]. It has been shown that the levels of RBFOX2 are significantly decreased in epithelial ovarian cancer [106]. Downregulation of RBFOX2 resulting in a shift to epithelial phenotype is also observed in breast cancers [106]. Ahuja et al. also found that the upregulation of RBFOX2 through TGF β signalling can also promote EMT by transcriptionally inhibiting ESRP1 in breast cancer [107]. More recently, Jbara et al. found that RBFOX2 plays an important role in focal adhesion formation and cytoskeleton organization, as well as functioning as a tumour suppressor in the metastatic process of pancreatic ductal cancer [108]. As EMT is generally linked to tumour progression and poor clinical outcome, several studies have indicated that EMT-linked alternative splicing patterns regulated by splicing regulators including RBFOX2 may be biomarkers of aggressive tumour types, especially in lung, breast, and colon cancer [36,109].

#### 4.2.3. Other Splicing Factors Involved in EMT

Expression of one of the serine/arginine (SR) proteins—**serine/arginine-rich splicing factor 1 (SRSF1)**—is upregulated in several cancers and is positively correlated with tumour growth and lymph node metastasis; additionally, it is negatively correlated with the sensitivity of cancer cells to anti-tumour therapy. The SRSF1 gene may regulate the occurrence of EMT in cancer cells by positively regulating the expression of the transcription factor SNAIL [110,111]. SRSF1 has also been found to regulate the expression of two isoforms of the MAPK pathway component MNK2—MNK2a and MNK2b—while the MAPK pathway mediates EMT in cancer [112,113]. Therefore, regulation of SR proteins is expected to provide new directions and new targets for the treatment of lung cancer through targeting AS in EMT.

**Heterogeneous nuclear ribonucleoproteins (hnRNPs)** are a large family of RNA-binding proteins (RBPs) involved in multiple processes of nucleic acid processing including alternative splicing. Increasing evidence shows that hnRNPs are involved in many types of cancers. They play a range of roles in resistance of apoptosis, angiogenesis, cell invasion, and EMT during cancer progression. Some observations indicate that hnRNP A2/B1 plays an critical role in alternative splicing of genes involved in cell migration, invasion, and EMT [114,115,116]. The increase in hnRNP A2/B1 is related to the poor prognosis of glioma patients, and its overexpression enables mouse fibroblasts to form high-grade sarcoma in nude mice [115]. TGF-β plays a crucial role in TGF-β-induced EMT and metastasis. hnRNPs are revealed to affect the TGF-β-modulated expression of EMT-specific proteins and EMT itself. Furthermore, knockdown of hnRNP K could notably reduce the mesenchymal phenotype induced by TGF-β in the non-epithelial lung cancer cell line A549 cells [117]. Other hnRNPs may also play a role in EMT. For example, hnRNP F regulates EMT in bladder cancer by intermediating the stabilization of Snail1 mRNA via binding to its 3′ UTR [118]. hnRNP F and H, in combination with other hnRNPs, are well-known to be functional in the regulation of AS [119]. hnRNP L can directly regulate the alternative splicing of many RNAs, and can also regulate the formation of circular RNAs through reverse splicing [120]. It is thus suggested that hnRNPs might be a novel and potential therapeutic target through regulating AS in EMT and as a biomarker for treatment response and prognostic evaluation in cancer management.

**Polypyrimidine tract-binding protein 1 (PTBP1)** is another hnRNP that sometimes acts as a splice factor. The expression of PTBP1 in the highly metastatic liver cancer cell line HCCLM3 is significantly increased, and the expression level of PTBP1 in liver cancer tissues is significantly higher than that in normal tissues. Overexpression of PTBP1 can significantly increase the migration and invasion of HCCLM3 cells, increase the expression of mesenchymal markers N-cadherin and vimentin, and promote the EMT process of liver cancer cells [121]. Studies have also found that PTBP1 can participate in the EMT of breast cancer cells and can affect the occurrence, invasion, and metastasis of tumours [122,123]. Therefore, PTBP1 can promote the migration and invasion of cancer cells by promoting the EMT, as well as targeting the signalling involved in regulating AS during EMT. This discovery provides a theoretical basis for PTBP1 as a new molecular target for breast cancer treatment.

**The neuro-oncological ventral antigen (NOVA) protein family, NOVA1 and NOVA2**, consists of RNA-binding proteins associated with alternative splicing and transport of some target mRNAs. Recent studies demonstrate that NOVA1 and NOVA2 promote EMT through β-catenin in breast cancer cells; NOVA proteins induced an increase in epithelial and decrease in mesenchymal markers expression by restoring β-catenin expression, proposing that NOVA proteins are potential targets in breast cancer management [124]. Qu et al. found that NOVA1 is a novel oncogene that is able to activate Wnt/β-catenin signalling, which is one of the critical signalling pathways that regulates EMT, with a consequent increase in the proliferation and invasion of non-small-cell lung cancer (NSCLC). Knockdown of NOVA1 has also been shown to suppress the proliferation, migration, and invasion of NSCLC cells [125]. Therefore, NOVA1 might be a potential target in NSCLC and breast cancer diagnosis and therapy.

**Muscleblind-like 1 (MBNL1)**, a gene implicated in myotonic dystrophy that also acts as a splice factor, is identified as a suppressor of multiorgan breast cancer metastasis. MBNL1 suppresses cell invasiveness by improving the stability of these genes’ transcripts. Consistently, increased MBNL1 expression is related to reduced breast tumour metastasis [126]. Another study demonstrates that MBNL1 destabilizes SNAIL transcripts and suppresses the EMT of colorectal cancer cells through the SNAIL/E-cadherin axis in vitro [127]. In addition, new discoveries about these genes will reveal the transcriptome during EMT/MET in more detail and deliver new potential therapeutic targets for the management of human cancers.

## 5. Conclusions

EMT is regulated by many factors in cancer progression. During dynamic change between EMT and MET, tumour cells can migrate to neighbouring organs or metastasize at a distance and can also develop resistance to traditional chemotherapy. However, studies have shown that EMT is not an irreversible process. Reversing or inhibiting EMT may be an effective way to inhibit tumour cell migration or distant metastasis. Among different levels of modulation of EMT, alternative splicing plays an important role. The in-depth study of AS and EMT during cancer progression not only helps to better understand the occurrence and regulation of EMT in cancer, but also provides a new direction for understanding the mechanism of tumour metastasis and recurrence and provides new treatment avenues. The development of new anti-tumour drugs targeting AS and EMT will hopefully become a new research direction. Multiple genes, splicing factors, and biological processes involved in AS in EMT during cancer progression provide various targets for novel therapeutic strategies in cancer management (summarized in Table 1).

## Figures and Tables

**Figure 1 genes-14-02001-f001:**
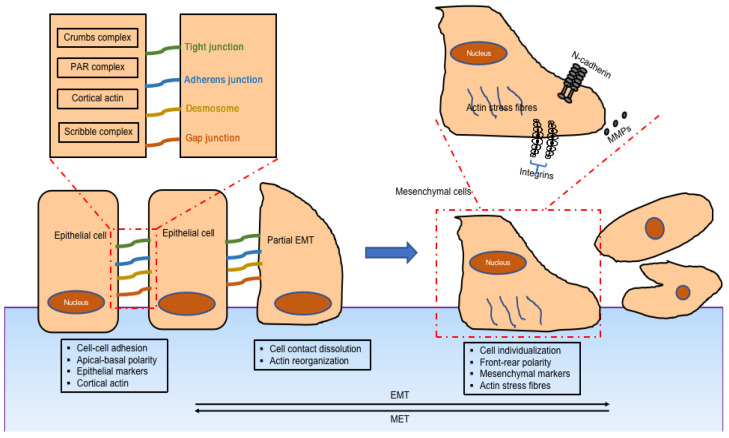
Mechanism of epithelial-mesenchymal transition: change of morphology, change of cell markers, and change of function when epithelial cells transition into mesenchymal ones [8].

**Figure 2 genes-14-02001-f002:**
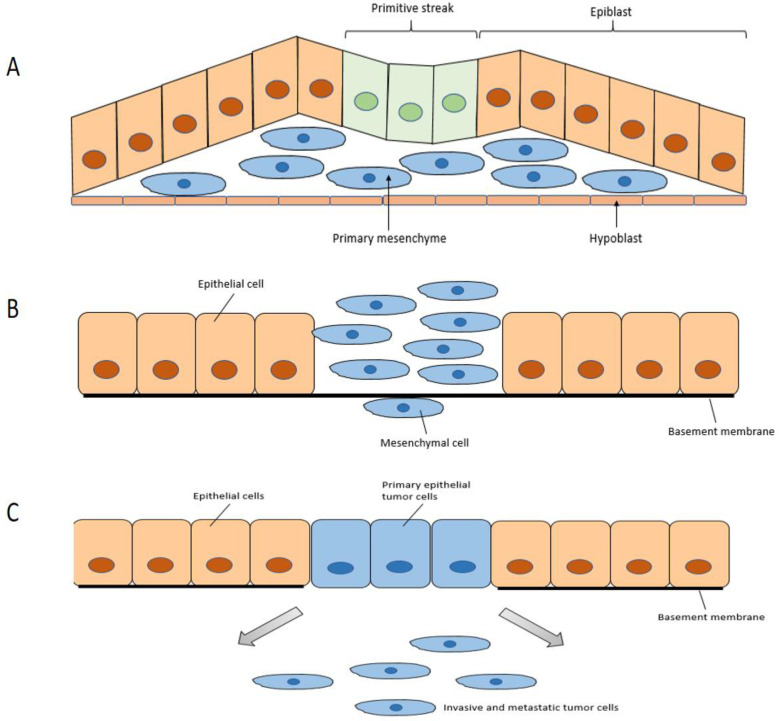
Three types of EMT: (**A**) Type 1 is associated with embryogenesis; (**B**) Type 2 is associated with wound healing and fibrosis; and (**C**) Type 3 is linked to the invasion and metastasis of tumours.

**Figure 3 genes-14-02001-f003:**
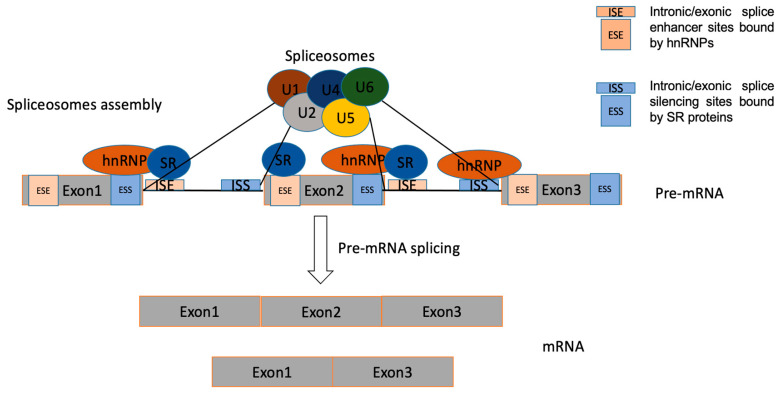
The process of splicing. The interaction of splicing factors with binding elements can either promote or inhibit spliceosome assembly to manipulate the use of 5′ or 3′ splice sites. The interaction of serine-rich/arginine (SR) proteins or hnRNPs with exonic splicing enhancers/silencers (ESE/ESS) or intronic splicing enhancers/silencers (ISE/ISS) promotes or inhibits the splice site utilization [28].

**Figure 4 genes-14-02001-f004:**
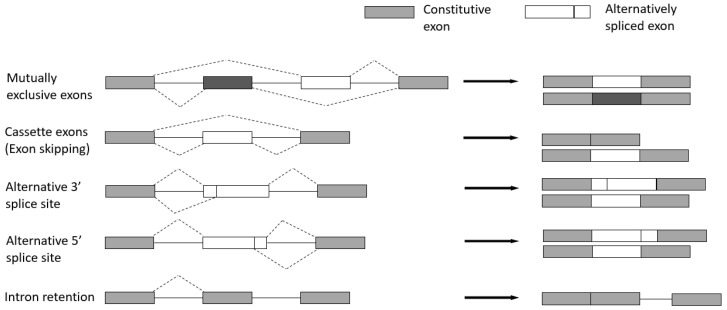
Classification of alternative splicing mechanisms: mutually exclusive exons, cassette exons, alternative 3′ and 5′ splice site, and intron retention.

**Table 1 genes-14-02001-t001:** Alternative splicing in EMT in cancer.

Models of Regulating EMT	Genes	Functions in EMT and Cancer
Specific alternative splicing patterns	FGFR2	FGFR2 IIIb: epithelial phenotype; FGFR2 IIIc: mesenchymal phenotype
	RON	RONΔ165: induce EMT through activation of Akt/PKB signalling; RONΔ160: triggers structural alterations that induce cellular transformation and tumour growth
	CD44	CD44v: higher expression in epithelial cells, suppress EMT; CD44s: increased in high-grade human breast tumours and was associated with N-cadherin increase
	CTNND1	p120-1: mostly expressed in mesenchymal cells; p120-3: predominant in epithelial cells, may revise EMT and inhibit tumour growth
	ARHGEF11	promote tumour metastasis and proliferation, and EMT of hepatocellular carcinoma by activating β-catenin; ARHGEF11 exon 38(+) correlated to the malignant phenotype in breast tumours
	CCND1	cyclin D1b: alone can induce cellular transformation, and be correlated with cancer progression and poor prognosis in prostate, colon, colorectal, and urinary bladder cancer
	AR	AR3: inducing EMT in prostate cancer
	ZAP	ZAPL and ZAPS play different roles at different stage of EMT
	MBNL1	MBNL1 Δex7: tumour suppressor that cancer cells tend to downregulate in presence of MBNL1 isoforms containing exon 7
Splicing regulators	ESRP1/2	regulate splicing events of ~200 genes, including FGFR2, maintain splicing towards epithelial phenotype
	RBFOX2	regulating splicing, drives a mesenchymal tissue-specific splicing program in EMT in cancer
	SR proteins	SRSF1 may regulate the occurrence of EMT in cancer cells by positively regulating the expression of the transcription factor SNAIL
	hnRNPs	play a role in resistance of apoptosis, angiogenesis, cell invasion, and EMT during cancer progression; PTBP1 can promote the migration and invasion of cancer cells by promoting the EMT
	NOVA	NOVA1 and NOVA2 promote EMT through increase in epithelial and decrease in mesenchymal markers expression by restoring β-catenin expression
	MBNL1	destabilizes SNAIL transcripts and suppresses the EMT of colorectal cancer cells through the SNAIL/E-cadherin axis in vitro

## Data Availability

Not applicable.

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
