# Peer review of "Regulation of Epithelial-Mesenchymal Transitions by Alternative Splicing: Potential New Area for Cancer Therapeutics"

_genes, 2023, doi:10.3390/genes14112001_

Round 1
Reviewer 1 Report
Comments and Suggestions for Authors
This manuscript reviews the role of alternative splicing in EMT with mainly an eye in cancer, by focusing on the functional implications of specific splicing events and regulators. Overall the manuscript should be useful to a variety of readers, however it falls short of its purpose because of its extremely descriptive nature. As it is now, this manuscript only describes the EMT/cancer data but brings no insights into the splicing regulatory mechanisms, which should be accommodated as much as space allows (I am not sure how much room for new text there is, editors to assess). Hence I am recommending few major changes.
Major comments:
1. In section 3.1, authors should elaborate on the regulatory mechanisms of these splicing events. For instance, RON has been studied by the Biamonti lab both from an EMT/cancer and mechanistic focus, however not a single of his papers was cited or discussed. In turn, there are many papers on the regulation of CD44 alternative splicing, among which the work of Chonghui Cheng is most relevant (only one paper cited). In particular, for these events it would be important to know the regulatory sequences, factors, relation to other processes like transcription on co-transcriptional splicing, etc.
2. Likewise, for section 3.2, the regulatory mechanisms of these splicing factors in EMT should be discussed. For instance, whether the splicing factors act as activators or repressors, their binding sites identified on targets, mode of action, etc.
3. A graphical summary at the end would be useful, with the splicing events and factors earlier discussed.
Minor comments:
4. In the Intro, it should be emphasized that EMT is relevant for carcinomas which are the cancers from epithelial origin, the majority but not all cancers.
5. Page 3 line 82, ‘occluding’ should read ‘occludin’ because it is a protein.
6. Figure 4, the dashed lines for alternative 5’ and 3’ splice sites are depicted at the wrong place (this is a very common figure design, it will easy to find through google images where these lines should be).
7. Correct or common nomenclature should be adopted: exonic/intronic splicing enhancers/silencers (rather than splice). Splicing factors rather than splice factors.
8. More splicing events can be found than the ones in section 3.1.5. For instance, last year the antiviral protein ZAP was shown to differentially affect EMT depending on the splice isoform. This is important to show that the connection between alternative splicing and EMT spans beyond the genes already documented to regulate this process, to include other unsuspected genes.
9. Gene/protein names should be used consistently: PTB should be referred to as PTBP1 or PTBP2. By the way, this factor is also an hnRNP so it should be mentioned on page 3.2.3.
10. Also on section 3.2.3, I suggest to remove the text and citations about hnRNPE1 if this factor regulates EMT via processes other than splicing.
Comments on the Quality of English Language11. Last, the manuscript will benefit from grammar curation.
Reviewer 2 Report
Comments and Suggestions for Authors
In this article by Li et al, the authors explain mechanisms of epithelial-mesenchymal transition (EMT) and alternative splicing (AS), as well as describe its regulatory factors. The authors specifically focus on the importance of alternative splicing in EMT during cancer development. They further discuss potentials of targeting alternatively spliced transcripts and splicing factors involved in EMT as a strategy for cancer treatment. This is a significant topic, especially with respect to the therapeutic applications.
The authors should address the following points prior to publication:
1) The manuscript is missing the Introduction section.
2) The review would greatly benefit if the authors could summarize, either in a Table or a Figure, candidate target genes discussed in chapter 3.
Round 2
Reviewer 1 Report
Comments and Suggestions for Authors
All ok now